# Associations between Serum Saturated Fatty Acids Content and Mortality in Dialysis Patients

**DOI:** 10.3390/jcm11175051

**Published:** 2022-08-28

**Authors:** Malgorzata Sikorska-Wisniewska, Adriana Mika, Tomasz Sledzinski, Michal Chmielewski

**Affiliations:** 1Department of Nephrology, Transplantology and Internal Medicine, Medical University of Gdansk, 80-210 Gdansk, Poland; 2Department of Pharmaceutical Biochemistry, Faculty of Pharmacy, Medical University of Gdansk, 80-210 Gdansk, Poland; 3Department of Environmental Analysis, Faculty of Chemistry, University of Gdansk, 80-309 Gdansk, Poland

**Keywords:** saturated fatty acids, lipids, dialysis, chronic kidney disease

## Abstract

Background: Cardiovascular mortality in dialysis population remains very high. Saturated fatty acids (SFA) contribute to atherosclerosis and to cardiovascular risk. Aim: The aim of this study was to evaluate the relationship between mortality in dialysis patients and the serum SFA content. Methods: Survival of 54 patients on dialysis was assessed. A total of 21 SFA from patients’ sera were measured by gas chromatography-mass spectrometry (GC-MS). Diet was assessed by food frequency questionnaire FFQ-6. The SFA content is presented as fatty acid proportion (%). Results: During the observation time (median 66 months) 22 patients died. There was a significant relationship between elevated SFA (above SFA mean) and mortality (log-rank 3.13; *p* = 0.0017). Moreover, patients who ingested foods rich in SFA, according to FFQ-6, had a higher mortality risk (log-rank 2.24; *p* = 0.03). The hazard ratio for mortality associated with increased SFA content equalled 4.47 (1.63–12.26). Addition of age and inflammation (hsCRP > 5 mg/L) into the Cox model did not modify this relationship. However, SFA content turned out to be significantly higher in patients with diabetes mellitus and cardiovascular disease, as compared to patients free from these co-morbidities. Their addition to the model attenuated the relationship between SFA and mortality, making it statistically insignificant. Conclusion: The serum content of SFA turned out to be a strong predictor of mortality in dialysis patients. However, given the significant associations between SFA, DM, and CVD, interventional studies with controlled SFA intake are needed to evaluate the causal links between SFA, co-morbidities and survival.

## 1. Introduction

The prevalence of chronic kidney disease (CKD) is estimated at a rate as high as 6–13% of the general population [1,2]. Moreover, it is still increasing, owing to the increased proportion of elderly individuals and patients diagnosed with diseases that contribute to CKD such as obesity, hypertension, diabetes mellitus (DM), and cardiovascular disease (CVD). Mortality rates of CKD patients are higher than in people free from kidney diseases, and they increase with the disease progression [3]. In patients with end-stage renal disease (ESRD) treated with dialysis, mortality risk is several times higher than in the general population with CVD being the major cause of premature death.

In the general population, lipid disorders are among the most acknowledged risk factors for CVD. In dialysis patients, atherosclerotic dyslipidaemia is prevalent. Hypertriglyceridemia and low-HDL cholesterol concentration are the most typical and potentially deleterious lipid abnormalities in the course of CKD [4]. However, our previous studies, among others, have also shown profound alterations in other lipid classes including significant disturbances in the serum profile of fatty acid content [5].

Fatty acids, carboxylic acids with an aliphatic chain, either saturated or unsaturated, play numerous crucial functions in the organism mainly as energy sources and membrane constituents. Saturated fatty acids (SFA) have only single bonds in their chains. They are mainly obtained through dietary intake of animal fats. In the general population, SFA have been linked to increased cardiovascular risk. Leading to an increase in low density lipoprotein (LDL) cholesterol concentration, they are thought to promote atherosclerosis, and hence, cardiovascular disease and mortality risk [6,7]. However, current studies in this area demonstrate discrepant results, generating contrary guidelines and opinions from firm recommendations to control SFA intake to a total neglection of their role in CVD progression [8,9]. Data in the CKD population are scarce, although there are studies demonstrating direct associations between the total serum SFA level and the risk of sudden cardiac death, perhaps in a mechanism that includes mitochondrial and cell damage triggered by SFA accumulation [10,11].

The aim of this study was to evaluate the relationship between mortality in dialysis patients and the serum content of saturated fatty acids (SFA).

## 2. Materials and Methods

The study was performed based on a group of 54 ESRD patients, treated with dialysis in a single university-based dialysis centre. Exclusion criteria were as follows: intake of fatty acids supplements, acute inflammatory state, and unwillingness to participate in the study. Half of the group (*n* = 27) was undergoing hemodialysis (HD) treatment, while the other half was on peritoneal dialysis (PD). Presence of diabetes mellitus (DM) and CVD was determined on the basis of patients’ medical charts.

Blood was collected prior to an HD session in HD patients and during a routine check-up visit in PD subjects. Serum was stored at −80 °C, until analysed. A total of 21 SFA from patients’ sera were measured by gas chromatography-mass spectrometry (GC-MS). Total lipids were extracted from whole serum samples in a chloroform–methanol mixture (2:1, *v*/*v*) following the method by Folch et al. [12], as described previously [5]. The SFA content is presented as fatty acid proportion (%). Circulating levels of high-sensitivity C-reactive protein (CRP), albumin, total and HDL cholesterol as well as triglycerides were analysed using certified methods at the central university laboratory.

Diet was assessed by food frequency questionnaire (FFQ-6), the most common dietary assessment tool used in large epidemiologic studies of diet and health and validated for the national population [13].

Protocol of the study received approval from the Local Bioethics Committee at the Medical University of Gdansk (protocol no. NKEBN/614/2013-2014) and informed consents were obtained from all the patients.

Results are expressed as mean and standard deviation or median and interquartile range, as appropriate. The assumption of normality was verified with the Kolmogorov–Smirnov test. A *p* value < 0.05 was considered to be statistically significant. Comparisons between two groups were assessed with a Student’s *t*-test, or a Mann–Whitney test, as appropriate. To assess correlations among the evaluated variables, Pearson’s correlation coefficient (r) was used. Survival analyses were made with the Kaplan–Meier survival curve and the Cox proportional hazard model, presenting data as hazard ratio (HR; 95% confidence intervals (CI)). Statistical processing of the results was performed with the use of the statistical software Statistica PL version 13.0 (StatSoft, Krakow, Poland).

## 3. Results

Out of the 54 patients under study, 12 had been diagnosed with DM, and 26 with CVD. The median of the observation time was 66 months (range 2–76 months). During the observation, 22 patients died. Kaplan–Meier analysis revealed a significant relationship between elevated SFA (above SFA mean) and mortality (log-rank 3.13; *p* = 0.0017) (Figure 1). In the Cox regression model, the hazard ratio for mortality associated with increased SFA content equalled 4.47 (1.63–12.26). Moreover, patients who ingested foods rich in SFA, according to FFQ-6, had a higher mortality risk (log-rank 2.24; *p* = 0.03) (Figure 2). However, as seen in Table 1, containing the general characteristics of the studied patients divided into groups according to SFA mean, patients with high SFA content turned out to be slightly older, more inflamed, and had a higher prevalence of DM and CVD (Table 1). Indeed, SFA content was significantly higher in diabetic patients, as compared to patients free from DM (35.2 ± 1.6 vs. 33.3 ± 2.8; *p* = 0.03), and was elevated in patients with CVD when compared to patients without it (34.9 ± 1.8 vs. 32.6 ± 2.9; *p* < 0.001). In Cox regression, the addition of age and inflammation (hsCRP > 5 mg/L) into the model did not modify the relationship between high SFA and mortality. However, addition of DM and CVD attenuated this relationship, making it statistically insignificant; HR 2.47 (0.78–7.71). According to food questionnaires, SFA content was associated with ingestion of food rich in saturated fat, although these relationships did not reach statistical significance. No associations between SFA and lipoprotein subclasses or with total cholesterol were observed. Sub-group analyses in patients stratified according to the use of HMG-CoA inhibitors revealed comparable results (Appendix A). There were also no significant differences in the analysed variables between HD and PD patients (Appendix A).

## 4. Discussion

In the present study, the serum content of SFA in dialysis patients turned out to be a strong predictor of mortality. However, since it was heavily dependent on the presence of DM and CVD, including these co-morbidities into the analysis attenuated the associations between SFA and outcome to a statistically insignificant level. Moreover, patients who ingested foods rich in SFA, according to FFQ-6, had a higher mortality risk in our analysis. Similarly, this association lost its significance after accounting for the above confounders.

In the general population, SFA intake and their content is generally believed to promote atherosclerosis, and in consequence, the cardiovascular risk [8]. The major mechanism probably includes the impact of SFA intake and content on increasing the total and LDL cholesterol level [14]. However, no clear associations have been observed between high content/intake of SFAs and the risk of atherosclerotic progression, and most of our understanding of their potential detrimental role is derived from observational studies, which are subject to numerous confounders as demonstrated in our study.

Cardiovascular disease is responsible for the largest proportion of deaths worldwide, and fatty acid profile is a potentially modifiable risk factor for the development of CVD, e.g., through dietary modifications; hence studies evaluating this issue seem of utmost clinical importance. Most public health dietary guidelines recommend limiting SFA intake [15,16,17]. These recommendations are based on numerous studies, mainly observational, demonstrating cardiovascular benefits associated with reducing SFA intake and/or replacing it with dietary polyunsaturated fatty acids (PUFA). For instance, a meta-analysis of eight clinical studies on replacing SFA with PUFA (combined *n* = 13,614 participants) estimated a CVD risk reduction of about 10% for each 5% energy replacement; an effect comparable to that predicted from the effects of the intervention on TC:HDL cholesterol ratio [18]. This may be especially important in dialysis patients, since our previous study showed decreased PUFA content in this group [19].

However, there are also opinions neglecting the role of SFA in CVD risk. A meta-analysis summarizing data related to SFA and CVD from 21 prospective epidemiologic studies (combined *n* = 347,747 participants) did not demonstrate significant associations between the intake of SFA and the risk of chronic heart disease and/or stroke [20]. These findings were consistent with yet another systematic review that showed a nonsignificant association of SFA with CVD [21]. It might be that SFA are most commonly replaced with carbohydrates, hence the lack of association between SFA and the risk of CVD can be interpreted as the lack of benefit of substitution of carbohydrates for saturated fat. Indeed, a pooled analysis of 11 cohort studies (combined *n* = 344,696 persons) demonstrated that replacement of SFA with polyunsaturated fat, but not carbohydrate or monounsaturated fat, was associated with decreased CVD risk [22].

Regardless of the mechanisms, the discrepant results of the above studies and meta-analyses result in contrary guidelines and opinions, either recommending reduction in SFA intake or neglecting such an approach [8,9].

In dialysis patients, the data on SFA with regard to CVD risk are scarce. Individual studies demonstrate some relations between serum SFA content and mortality in this patient population. In an analysis by Friedman et al. [11], a 0.1% increase in total serum SFA levels was associated with a 1% increase in the odds of sudden cardiac death of HD patients. The authors linked their observations to pro-arrhythmic potential of SFA, observed in some experimental studies [23,24]. The significant association between mortality and SFA intake, found in our study, supports (to some extent) the importance of SFA intake in the dialysis patient population.

Our results show that observational studies are laden with the influence of confounders. The content of SFA turned out to be significantly higher in diabetic patients, as compared to subjects free from DM. This obviously influenced the associations between SFA and outcome, as DM patients are characterized by substantially increased mortality risk, observed also in our analysis. It is difficult to speculate on the potential mechanisms of increased SFA in diabetic patients. It might be that their diet, with limited carbohydrate intake, is automatically more abundant in lipids, including SFA. However, there are studies actually linking circulating SFA to the incidence of type 2 DM [25,26,27]. Diet rich in SFA is typically associated with increased risk of weight gain, a risk factor for type 2 DM development. Moreover, SFA and their derivatives, ceramides, have been shown to induce insulin resistance [28,29].

Similarly, in our analysis, serum SFA content was increased in patients who had been diagnosed with CVD. This severe co-morbidity is constantly associated with impaired survival. This association was also evident in our study, as CVD patients presented with a significantly worse mortality risk in survival analysis, in comparison to dialysis subjects without this co-morbidity. In the general population, the links between dyslipidemia and CVD are even more acknowledged than in the case of DM. SFA, through their impact on the concentration of LDL cholesterol and on the structure of LDL particles, promote atherosclerosis, and hence CVD [6].

Obviously, to determine whether the links among SFA, DM, CVD and patient outcome are causative and to track potential mechanisms responsible for the above associations, prospective interventional studies are needed. In the general population, such studies are limited by a reliance on nutritional assessment methods of varying reliability, and the assumption that diets remain constant over the study period. In dialysis patients, additional difficulties include other, disease-associated dietary restrictions, and the risk of malnutrition, or protein-energy wasting. Indeed, additional mechanisms and/or non-causative relationships between SFA and mortality include their associations with sarcopenia. This deleterious complication is common in dialysis population and has a strong impact on patient survival [30]. Reduction in SFA intake for the benefit of polyunsaturated fatty acids (PUFA) consumption was shown to decrease the risk of sarcopenia [31]. It must be stressed, however, that in our analysis we have not found any associations between SFA and body mass index (BMI).

Substantial limitations of the present study need to be addressed. The major one is the small number of patients, additionally treated in only one dialysis unit. However, in our opinion even with these numbers the results are convincing. The association between SFA content/intake and mortality is evident, as shown in Kaplan–Meier analyses. However, as the study demonstrates, the patient outcome is probably due to other factors, such as co-morbidities, not to the detrimental impact of SFA themselves. Another potential limititation might be combining patients treated with different dialysis methods (PD and HD), as well as patients treated with lipid-lowering therapy. Patients treated with PD are characterized by different CV risk factors than HD patients. For instance, they have a tendency for higher cholesterol concentrations because of the dialysis-associated glucose load, while HD patients are subject to lower hemoglobin concentrations because of blood losses during the dialysis procedure. Still, the CV mortality and the overall outcome are comparable with these two dialysis methods, as are the baseline characteristics of patients in our study who were stratified according to their treatment modality (Appendix A). Similarly, there were no significant differences in patients stratified according the the use of lipid-lowering medications (Appendix A). Furthermore, the FFQ-6, although validated for the national population, has not been verified in dialysis patients. Finally, the observational character of the study precludes analysis of causative relationships between SFA and outcomes and leaves us only with hypotheses.

## 5. Conclusions

In conclusion, in our cohort of patients undergoing dialysis a strong relationship between serum SFA content, as well as SFA intake and survival, has been demonstrated. This association turned out to be significantly dependent on the co-existing illnesses that themselves constitute strong risk factors for poor outcomes. Whether SFA contributes to these diseases or acts just as innocent by-standers, can be determined solely through interventional studies.

## Figures and Tables

**Figure 1 jcm-11-05051-f001:**
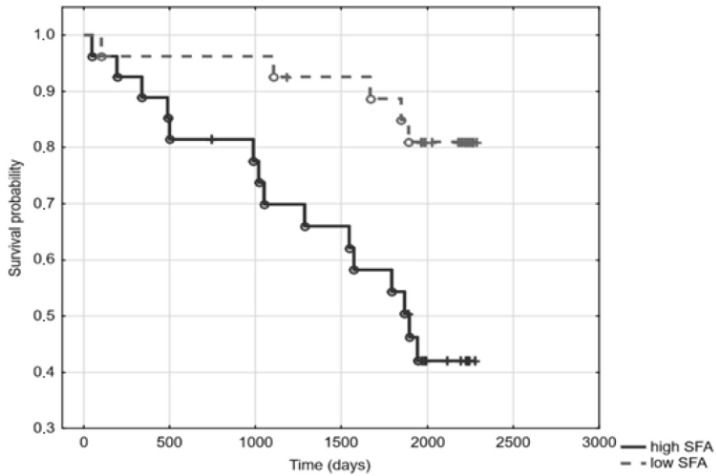
Survival curves of dialysis patients according to their saturated fatty acids (SFA) content.

**Figure 2 jcm-11-05051-f002:**
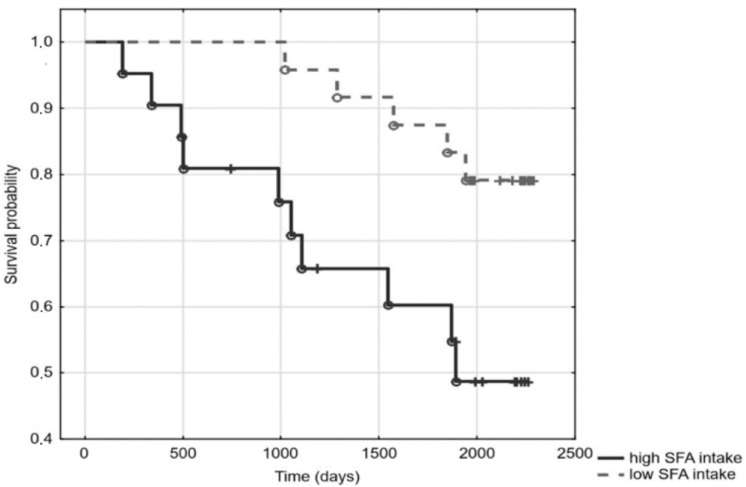
Survival curves of dialysis patients according to their saturated fatty acids (SFA) intake.

**Table 1 jcm-11-05051-t001:** Baseline characteristics of the studied dialysis patients stratified according to their saturated fatty acids (SFA) content; CRP—high sensitivity C-reactive protein, DM—diabetes mellitus, CVD—cardio-vascular disease, TC—total cholesterol, HDL—high density lipoprotein cholesterol, LDL—low density lipoprotein cholesterol, TG—triglycerides.

	Low SFA (*n* = 27)	High SFA (*n* = 27)	*p*-Value
Age (years)	53.8 ± 15.7	59.9 ± 9.5	0.09
CRP (mg/L)	2.94 (1.17–7.81)	6.80 (1.64–9.71)	0.06
DM (%)	10.7	32.1	<0.01
CVD (%)	25.0	71.4	<0.01
Albumin (g/L)	32.4 ± 4.0	31.6 ± 4.2	0.45
TC (mg/dL)	203.0 (167.0–247.0)	169.0 (151.0–203.5)	0.12
HDL (mg/dL)	38.5 (34.0–47.0)	37.0 (32.0–43.5)	0.43
LDL (mg/dL)	115.0 (99.0–117.0)	92.5 (72.0–125.0)	0.05
TG (mg/dL)	192.0 (108.5–197.0)	178.5 (121.5–233.0)	0.36

## Data Availability

Data is contained within the article or Appendix A.

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
