# Peer review of "Associations between Serum Saturated Fatty Acids Content and Mortality in Dialysis Patients"

_jcm, 2022, doi:10.3390/jcm11175051_

Round 1

Reviewer 1 Report

Sikorska-Wiśniewska M and collaborators evaluated the role of serum saturated fatty acid on the mortality of dialysis patients. The authors selected 54 patients on extracorporeal and peritoneal dialysis and followed them up to approximately 66 months. They measured a total of 21 SFA from patients' sera  by gas chromatography-mass spectrometry (GC-MS). Diet was assessed by the food frequency questionnaire FFQ-6. The results demonstrated that elevated SFA levels were associated with increased mortality. However, SFAs were also associated with diabetes and cardiovascular disease so adding these to the Cox model nullified significance. The work does not demonstrate anything particularly new given the known  role that SFAs have on atherosclerosis processes. The sample size is too small for a study of this type and the conclusions are irrelevant since the association between increased  SFA levels and mortality was not significant when diabetes and cardiovascular disease were also included in the statistical model

Author Response

Response to the comments made by Reviewer 1

This is, obviously, an unfavorable review of our work. We have to admit that it rightly emphasizes the major caveats of the analysis. The studied group is indeed small. However, in our opinion, even with these numbers, the results are convincing. The association between SFA content/intake and mortality is apparently evident, as shown in Kaplan-Meier analyses. However, as the study demonstrates, the patient outcome is probably due to other factors, as co-morbidities, not to the detrimental impact of SFA themselves. The Reviewer states that the role of SFAs in atherosclerosis process in known. Still, no clear associations were observed between high content/intake of SFAs and the risk of atherosclerotic progression, and most of our understanding of their potential detrimental role is derived from observational studies, which are subject to numerous confounders, as demonstrated in our study. This thesis has now been underlined in our manuscript.

The Reviewer also states that ‘the conclusions are irrelevant since the association between increased SFA levels and mortality was not significant’. We would like to oppose to this approach. We believe that reporting and publishing findings with only statistically significant outcomes leads to a phenomenon termed a publication bias. In our opinion, every interesting finding, whether significant or not, is of equal scientific importance. 

We hope that with these responses we were able to diminish, at least to some extent, the reviewer’s unfavorable opinion about our study.

Reviewer 2 Report

The paper is of interest, however it suffers for many and relevant drawbakcs.

First of all, as yet Authors undrelinned, the population enrolled is quite small.

Moreover, HD and PD patients are put togheter, however these two dialysis treatment show different CV risk, this may be a possible bias. 

Another point that reserve clarification is the role of LDL (see Table n. 1) , significantly lower in patients with higher SFA. How explain these findings, the Authors are requested to comment these data. 

Last but not least, the role of current therapies is not clear.. The Authors quote that "subgroup analysis  according to the use of HMG-CoA inhibitors revealed comparable results", but no data are shown for all lipid lowering therapies. 

Author Response

Response to the comments made by Reviewer 2

We would like to thank the Reviewer for the interest in our study and the valuable remarks.

Comment 1: the population enrolled is quite small.

Response: Indeed, this is an important limitation of our analysis, as underlined in the Discussion.  However, in our opinion, even with these numbers, the results are convincing. The association between SFA content/intake and mortality is apparently evident, as shown in Kaplan-Meier analyses. However, as the study demonstrates, the patient outcome is probably due to other factors, as co-morbidities, not to the detrimental impact of SFA themselves.

Comment 2: HD and PD patients are put together, however these two dialysis treatment show different CV risk, this may be a possible bias.

Response: We agree that these two methods of RRT are associated with different CV risk factors. For instance, PD patients can have higher cholesterol concentrations due to dialysis-associated glucose load. Indeed, in our analysis, cholesterol tended to be higher in PD than in HD patients. This data is provided below, and also added to the current version of the manuscript as supplementary material. However, we cannot agree that these two methods show different CV risk. There are no studies convincingly showing superiority of one method over the other. If that was the case, we wouldn’t offer the patients the choice, we would treat them all with the method associated with better CV outcome. In general, these are all end-stage kidney disease patients treated with dialysis. They all vary in their sex, age, primary kidney disease, co-morbidities AND their dialysis method. The only common denominator is their CKD stage5D. We have referred to that issue in the present version of the manuscript.

Comment 3 and 4: Another point that reserve clarification is the role of LDL (see Table n. 1) , significantly lower in patients with higher SFA. How explain these findings, the Authors are requested to comment these data. Last but not least, the role of current therapies is not clear. The Authors quote that "subgroup analysis according to the use of HMG-CoA inhibitors revealed comparable results", but no data are shown for all lipid lowering therapies. 

Response: We would like to respond to these two comments (on LDLs, and on lipid lowering therapy) together, as, in our opinion, they are related to each other. Indeed, the high SFA group is characterized by a lower LDL-cholesterol level. One of the probable explanations is the impact of statin therapy on LDL-cholesterol. The high SFA group is characterized by higher CVD and DM prevalence. These patients more often receive statin therapy. Indeed, this is also the case in our analysis. Therefore, statins were used by a slightly higher percentage of patients in the high SFA group (63% vs. 44%). And patients on statins have had lower LDL cholesterol concentrations, as shown in the table below. This data on lipid-lowering medications has now been added to the manuscript as supplementary material.

PD (n=27)

HD (n=27)

p-value

Age (years)

54.0 ± 11.6

59.7 ± 13.2

0.09

CRP (mg/l)

3.4 (1.4-7.8)

6.8 (1.6-12.3)

0.22

Albumin (g/l)

31.9 ± 4.7

32.4 ± 3.4

0.60

TC (mg/dl)

202 (167-245)

177 (146-203)

0.07

HDL (mg/dl)

38 (31-42)

37 (35-49)

0.40

LDL (mg/dl)

132 (93-167)

106 (73-129)

0.07

TG (mg/dl)

178 (127-228)

149 (74-230)

0.18

SFA (%)

34.0 ± 2.1

33.4 ± 3.0

0.22

statin (n=29)

no statin (n=25)

p-value

Age (years)

59.1 ± 11.4

53.8 ± 12.8

0.11

CRP (mg/l)

5.8 (1.4-12.3)

3.6 (1.8-7.8)

0.55

Albumin (g/l)

31.8 ± 4.1

32.4 ± 4.1

0.59

TC (mg/dl)

173 (140-208)

182 (156-249)

0.21

HDL (mg/dl)

39 (37-46)

36 (31-44)

0.08

LDL (mg/dl)

100 (73-130)

115 (93-182)

0.06

TG (mg/dl)

168 (78-208)

161 (113-285)

0.32

SFA (%)

34.1 ± 2.3

33.3 ± 2.8

0.13

Round 2

Reviewer 1 Report

I believe the authors responded appropriately to all the reviewers' comments